# Antimicrobial Resistance in Qatar: Prevalence and Trends before and Amidst the COVID-19 Pandemic

**DOI:** 10.3390/antibiotics13030203

**Published:** 2024-02-21

**Authors:** Hassan Al Mana, Hamad Abdel Hadi, Godwin Wilson, Muna A. Almaslamani, Sulieman H. Abu Jarir, Emad Ibrahim, Nahla O. Eltai

**Affiliations:** 1Biomedical Research Center, Qatar University, Doha P.O. Box 2713, Qatar; h.almana@qu.edu.qa (H.A.M.); eelmagboul@hamad.qa (E.I.); 2Communicable Diseases Centre, Hamad Medical Corporation, Doha P.O. Box 3050, Qatar; habdelhadi@hamad.qa (H.A.H.); malmaslamani@hamad.qa (M.A.A.); sabujarir@hamad.qa (S.H.A.J.); 3Laboratory Medicine and Pathology, Hamad Medical Corporation, Doha P.O. Box 3050, Qatar; gwilson@hamad.qa

**Keywords:** bacterial co-infection, COVID-19, hospital, AMR, Qatar

## Abstract

Antimicrobial resistance (AMR) is a global healthcare challenge with substantial morbidity, mortality, and management costs. During the COVID-19 pandemic, there was a documented increase in antimicrobial consumption, particularly for severe and critical cases, as well as noticeable travel and social restriction measures that might influenced the spectrum of AMR. To evaluate the problem, retrospective data were collected on bacterial infections and antimicrobial susceptibility patterns in Qatar before and after the pandemic from 1 January 2019 to 31 December 2021, covering 53,183 pathogens isolated from reported infection episodes. The findings revealed a significant resistance pattern for extended-spectrum beta-lactamase-producing *Enterobacteriaceae* (ESBL-EBC), carbapenem-resistant *Enterobacteriaceae* (CR-EBC), and carbapenem-resistant *Pseudomonas aeruginosa* (CRPA), ciprofloxacin-resistant *Salmonella* and methicillin-resistant *Staphylococcus aureus* (MRSA). For correlation with social restrictions, ESBL-EBC and MRSA were positively correlated with changing patterns of international travel (ρ = 0.71 and 0.67, respectively; *p* < 0.05), while CRPA was moderately correlated with the number of COVID-19 hospitalized patients (ρ = 0.49; *p* < 0.05). CREBC and CRPA respiratory infections were associated with hospitalized patients (OR: 3.08 and 2.00, respectively; *p* < 0.05). The findings emphasize the challenges experienced during the COVID-19 pandemic and links to international travel, which probably will influence the local epidemiology of AMR that needs further surveillance and control strategies.

## 1. Introduction

Antimicrobial resistance (AMR) manifests when microbes such as bacteria and viruses become insensitive to drugs previously used to treat them effectively. When the microbes become resistant to medication, the associated diseases become progressively challenging to treat, increase in severity, and result in an upsurge in mortality rates. The annual global mortality rates from drug-resistant microbes are projected to increase to approximately 10 million by 2050 [1]. Conditions most associated with bacterial resistance include tuberculosis, acquired immunodeficiency syndrome (AIDS), respiratory tract diseases, malaria, and nosocomial illnesses resulting from *Staphylococcus aureus* (*S. aureus*), *Acinetobacter*, *Escherichia coli* (*E. coli*), and *Klebsiella pneumoniae* (*K. pneumoniae*) [1]. AMR rates in bacteria continue to increase globally; for example, a study in the Philippines shows an overall increase in antimicrobial resistance from the 1990s to 2017 [2]. Furthermore, the WHO Global Antimicrobial Resistance and Use Surveillance System (GLASS) reported an increase in resistance rates across the globe of approximately 15% from 2017 to 2020 [3]. While an average increase of 5% per year may seem small, any increase is cause for concern as it can impact treatment and lead to mortality in many patients.

Nevertheless, a robust antimicrobial stewardship program can potentially lead to a reduction in rates, as shown by the decrease in AMR rates following the rollout of such a program in a hospital in Italy [4]. Large-scale disruptions to the healthcare system can significantly impact the epidemiology and dynamics of AMR. An understanding of the impact is essential in the development and implementation of effective, long-term stewardship programs.

Although COVID-19 is a viral disease caused by SARS-CoV-2, the chaos and disruption associated with it have a significant impact on antimicrobial- resistance [5]. Notably, the administration of antibiotics has increased since individuals hospitalized due to COVID-19 are likely to develop secondary nosocomial infections like pneumonia [5,6,7]. COVID-19 management involves immunosuppression, which may increase the incidence of opportunistic bacterial infections, making it necessary to administer antimicrobial drugs. A study noted that 67% of COVID-19 patients received antibiotics in their treatment to manage confirmed or suspected secondary bacterial infections [8]. Abu-Rub et al. (2021), Arastehfar et al. (2020), and Mędrzycka-Dąbrowska et al. (2021) acknowledge that the use of antibiotics during the COVID-19 period may have contributed to an increase in antimicrobial resistance [6,9,10].

Antibiotic administration is influenced by patient pressures, clinician knowledge, and laws and regulations [11]. In addition to the confusion associated with COVID-19, secondary bacterial infections and claims of the effectiveness of antibiotics such as azithromycin among COVID-19 patients have compounded the already worsening problem of antimicrobial resistance [12]. Additionally, a recent study conducted in Qatar shed light on imported resistance carriers and whether they may be associated with variations in antimicrobial resistance during the pandemic. Interestingly, the carriage rate for carbapenem-resistant *Enterobacteriaceae* (CR-EBC) has decreased since the imposition of travel restrictions following the COVID-19 outbreak [13].

Qatar is in the Arabian Peninsula, with a population of approximately 3 million. It has a universal healthcare system run by Hamad Medical Corporation (HMC). Notably, HMC serves the entire population, and all COVID-19 hospitalized patients were housed in their facilities. In our previous study, we investigated the changes in bacterial infection epidemiology throughout the pandemic using data from 1 January 2019 to 31 December 2021 and found an association between the number of international travelers and the number of infections as well as changes in the underlying bacterial populations [14]. Using the same dataset, this study aims to investigate the changes in AMR patterns across the period and compare COVID-19 hospitalized patients to non-hospitalized patients with bacterial infections.

## 2. Results

### 2.1. Data Characteristics

Of the 68,654 bacterial infections in the original dataset collected to investigate microbial dynamics during the pandemic [14], 53,183 (77.47%) are caused by the pathogens selected from the World Health Organization’s AMR priority list [15,16]. Table 1 shows a demographic summary of this subset of the data. The median patient age was 37, in line with the country demographics during the pandemic, and infections were more common in females and non-Qatari patients. The most common pathogens were *E. coli*, followed by *S. aureus* and *K. pneumoniae,* in line with the previously described epidemiology in the country (Table 2) [17]. The other *Enterobacteriaceae* group includes 34 species from 13 genera: *Citrobacter*, *Cronovbacter*, *Enterobacter*, *Escherichia*, *Klebsiella*, *Kluyvera*, *Leclercia*, *Lelliottia*, *Plesiomonas*, *Pluralibacter*, *Pseudecherichia*, *Raoultella*, *Shigella*. *E. coli*, *K. pneumoniae*, and *Salmonella* are also members of the family *Enterobacteriaceae*; however, the first two are responsible for over half the infections, and fluoroquinolone resistance in the latter is of particular concern. The number of infections caused by these organisms follows the same pattern described previously [14].

### 2.2. Antimicrobial Resistance Trends

Figure 1 shows the number of infections by key resistant pathogens stratified by the COVID-19 pandemic period. The key resistant pathogens are ESBL-producing Enterobacteriaceae (ESBL-EBC), carbapenem-resistant Enterobacteriaceae (CR-EBC), *Acinetobacter. baumannii* (CRAB), *P. aeruginosa* (CRPA), fluoroquinolone-resistant *Salmonella* (CIP-SAL), MRSA, and multidrug-resistant organisms (MDROs). There were statistically significant differences in the resistance rates across the pandemic periods for all these groups (*p* < 0.05), except for CRAB, which has a low prevalence, and CR-EBC.

Post-hoc tests were performed to pinpoint the observed differences in the proportion of antimicrobial-resistant infections. ESBL-EBC and CIP-SAL differed between the pre-restrictions and first gradual lifting periods, with 2% and 8% reduction in resistance rates, respectively (*p* < 0.005). MRSA differed between pre-restrictions and both the first gradual lifting (4% decrease; *p* < 0.005) and second restriction periods (6% decrease; *p* < 0.005). CRPA showed more variation, with differences between the second restrictions period and all of the pre-restrictions (6% increase; *p* < 0.0005), first gradual lifting (4% increase; *p* < 0.05), and second gradual lifting periods (5% decrease, *p* < 0.005). Interestingly, CRPA rates increased during the pandemic period. Lastly, MDROs differed between pre-restrictions and all other periods. The rate of MDRO infections decreased during the pandemic compared to prior, albeit to a small degree (~3–4%). No significant differences were observed for CRAB; however, recorded numbers are too small to make reliable observations.

Stratifying by infection type, significant differences were observed in UTIs for ESBL-EBC and MDROs, sterile site infections for CRPA and MDROs, and respiratory tract infections (RTIs) for CRPA, MRSA, and MDROs (Figure 2). While the changes in ESBL-EBC rates are statistically significant, they are small (2–3%). However, the others show more considerable differences in resistance rates, particularly in respiratory and sterile site infections. Notably, resistance rates increased in the first restriction period for CRPA, MRSA, and MDROs. On the other hand, sterile site infections increased during the first gradual lifting and second restrictions period for CRPA and decreased from MDROs (Figure 2). These results indicate changes in the patient population during the pandemic as more acute cases appear and need hospitalization. Additionally, existing patients have been hospitalized for an extended period and require immunosuppressive and antibiotic treatments, which may have promoted the growth of resistant organisms.

### 2.3. Antimicrobial Resistance and COVID-19 Hospitalization

Resistance rates were compared between infections during COVID-19 hospitalization and infections that occurred without COVID-19 hospitalization (regardless of the patient’s COVID-19 status). CR-EBC infections were 8.85 times more likely to appear in hospitalized patients (95% CI: 6.20–12.33; *p* < 0.01), CRPA infections were 3.26 times more likely (95% CI: 2.20–4.75; *p* < 0.01), and MDROs were 1.38 times more likely (95% CI: 1.14–1.67; *p* < 0.01). There was no significant difference between the two groups in ESBL-EBC, CRAB, CIP-SAL, and MRSA infections (Figure 3).

Stratifying by infection type showed that the odds of RTIs by CR-EBCs in COVID-19 hospitalized patients compared to non-hospitalized patients are 3.08 (95% CI: 1.92–4.80; *p* < 0.001). Similarly, MDRO and CRPA RTI are 1.73 times (95% CI: 1.34–2.22; *p* < 0.001) and 2.00 times (95% CI: 1.27–3.08; *p* < 0.005) more likely in hospitalized COVID-19 patients, respectively. As for sterile site infections, the only significant difference was with CR-EBCs, which are 3.08 times more likely in hospitalized patients (95% CI: 1.92–4.80; *p* < 0.001).

Spearman rank correlation was performed to measure the correlation between CRPA, ESBL-EBC, CR-EBC, and MRSA with the number of international visitors and hospitalized COVID-19 patients (the values for all spearman rank correlations are shown in Appendix A). The data is incomplete for the number of COVID-19 hospitalized patients; thus, the correlation was limited to the period where there is available data (14 May 2020–October 2021) which covers the latter half of the first restrictions period to half of the second gradual lifting period. There was a strong positive correlation between ESBL-EBC and MRSA with the number of international visitors (ρ = 0.71 and ρ = 0.67; *p* < 0.001), highlighting the possibility of community transmission and importation. On the other hand, there was a moderate positive correlation between CRPA and the number of hospitalized COVID-19 patients (ρ = 0.4; *p* = 0.03), highlighting the possibility of hospitalization as a risk factor. MDROs were also positively correlated with international visitors (ρ = 0.61; *p* < 0.001). However, this group encompasses the others in addition to other organisms and is strongly correlated with both ESBL-EBC and MRSA. 

## 3. Discussion

The historic COVID-19 pandemic caused a substantial impact on healthcare systems across the globe with noticeable clinical, social, and economic consequences [18,19]. In addition to the significant morbidity and mortality observed during the pandemic, several studies highlighted changing local and global disease epidemiology, including shifts in secondary bacterial infections, community and hospital-acquired infections (CAIs, HAIs), and AMR [20,21,22,23,24]. Steroids are commonly used in COVID-19 treatment, and their immune suppression side effect and prolonged hospitalization can leave patients susceptible to secondary bacterial infections and co-infections [24,25]. As such, COVID-19 patients are typically prophylactically prescribed antibiotics. Multiple studies reported increased antibiotic consumption during the pandemic, both in treatment and as prophylaxis [5,6,7]. The substantiated overconsumption of antimicrobials in COVID-19 patients probably exerts negative pressure on carrier/colonization state as well as active infections selecting resistant organisms [26].

When examining the effect of travel and social restrictions during the COVID-19 pandemic on the changing epidemiology of local microbial infections, we found a general decremental trend for bacterial infections in our previous study [14]. Building on this observation, the present inquiry focuses on bacterial AMR trends concerning the evolving pandemic examined through social and travel restrictions focusing on priority pathogens as listed by the WHO. These pathogens were responsible for 77.47% of infections in Qatar within the study period, with *E. coli*, *S. aureus*, and *K. pneumoniae* being the three most common causes of infection. Notably, Qataris, estimated to constitute approximately 13% of the population (https://worldpopulationreview.com/countries/qatar-population; accessed on 24 December 2023), accounted for 29% of the patient population during the study period. This disproportionate representation may be attributed to the healthy immigrant effect, as immigrants undergo comprehensive health checks before being granted visas and entry. Additionally, the pathogens considered for this study tend to be more common in nosocomial infections, and Qatari patients may be more likely to undergo extended hospitalization within the country.

There were statistically significant changes in the rates of ESBL-EBC, CR-EBC, CIP-SAL, MRSA, CRPA, and MDROs. Interestingly, CRAB did not change significantly, likely due to the small number of infection episodes with low resistance rates. The ESBL-EBC, CIP-SAL, and MRSA rates decreased substantially between pre-restrictions and the first gradual lifting period. These reductions could be credited to the COVID-19 restrictions that were set in place. For example, given that *Salmonella* infections are typically associated with travel to endemic areas as well as being primarily transmitted through foodborne routes that typically maximize through social gatherings [27], travel restrictions, social distancing, and other restriction measures together with increased awareness of hygiene practices during the pandemic likely hindered *Salmonella* transmission. A study in the Netherlands showed a decrease in the incidence of salmonellosis and a reduction in trimethoprim resistance but did not investigate ciprofloxacin resistance [28]. Additionally, ESBL-EBCs and MRSA infections were positively correlated with the number of international visitors (ρ = 0.71 and 0.67, respectively), which decreased significantly during the pandemic. Globally, international travel has been documented to be associated with ESBL carriage and infections [29,30]. Similarly, international travel and close contact are associated with MRSA infections [31,32].

Investigation of the resistance rates stratified by infection types showed a statistically significant yet small (2–3%) reduction in ESBL-EBC rates in UTIs. While there was an overall decrease in the rates of MRSA RTIs, there was a spike early in the pandemic, with 62% of *S. aureus* RTIs being resistant to methicillin. This peak coincides with the pandemic’s early stages and the imposition of COVID-19 measures. The number of new COVID-19 cases per month was the highest during the first restrictions period. While the data is incomplete for COVID-19 hospitalizations, there was a peak in hospitalizations in June 2020, indicating that the number was increasing prior (Appendix A). These factors, along with increased antibiotic consumption, weakening of the respiratory system, and immune suppression, may have contributed to this increase. The same factors may have also contributed to the increase of CRPA in RTIs during the same period.

Studies on the effects of the COVID-19 pandemic on AMR had varying results. Some showed increased incidence and outbreaks of MDROs, while others showed no changes or a decrease [33,34,35,36]. The conclusions differed based on the organisms studied and the underlying patient population. For example, one study showed an outbreak of vancomycin-resistant enterococci in a COVID-19 intensive-care unit (ICU) ward [34]. On the other hand, a survey on MDROs in France showed no difference in the rates [36]. This variation is reflected in our results, with pathogen and infection-type variations (Figure 2).

When comparing COVID-19 hospitalized patients with the general patient population, CR-EBCs, CRPAs, and MDROs were more likely in RTIs in COVID-19 patients (Odds ratios 3.08, 2.00, and 1.73, respectively). Furthermore, CRPA had a moderate positive correlation with the number of COVID-19 hospitalized patients (ρ = 0.49). While the results indicate that hospitalized COVID-19 patients are more likely to be diagnosed with MDRO infections such as CR-EBC and CRPA, it is essential to note that the present data compares COVID-19 hospitalized patients with the general patient population, and thus, a distinction cannot be made between an association with COVID-19 hospitalization or hospitalization in general. Nevertheless, COVID-19 hospitalization and treatment may put patients at a higher risk of these infections. A previously matched study from the same institutions highlighted prolonged hospital stay and mechanical ventilation, as well as previous exposure to antimicrobials, as a major risk factor for the acquisition of MRDOs in the critically ill [37]. Interestingly, a study on CR-EBC colonization in the pediatric population during the pandemic in Qatar found a positive correlation between colonization rates and the number of international travelers [13]. The absence of a correlation here suggests that the transmission of colonization and infections occurs under different mechanisms.

## 4. Materials and Methods

### 4.1. Setting

Qatar is located on the Arabian Peninsula, with a population of approximately 3 million, covered by a universal national healthcare system administrated through Hamad Medical Corporation (HMC) through 14 general and specialized healthcare facilities. Since HMC serves the entire population, and all records of COVID-19 outpatient and hospitalized cases are available through electronic records, epidemiological evaluations are nationally representative. A previous study by the same research group examined shifting patterns of bacterial and fungal infections before and throughout the pandemic, highlighting the association between COVID-19 disease, hospitalization, social and travel restrictive measures, and changes in pathogens epidemiology [14]. Using the same dataset and linking pathogens to antimicrobial susceptibilities (ASTs) to record resistance, the presented study aims to investigate the changes in AMR patterns across the period and compare COVID-19 hospitalized patients to non-hospitalized patients with bacterial infections. 

### 4.2. Data Management and Ethical Considerations

The study was approved by the Medical Research Centre (MRC) of Hamad Medical Corporation (HMC), which abides by local and international research standards (Protocol: MRC-02-21-949). The study also received approval from the Ethical Committee and Institution Review Board of the MRC after observing data management and sharing standards, including limited access to nominated primary investigators, data anonymity, and governance. All shared data had no traced patients’ identification. 

### 4.3. Dataset Specification and Pre-Processing

Retrospective data was collected on all bacterial infections from HMC between 1 January 2019 and 31 December 2021, as described previously [14]. This period includes 15 months before the first pandemic restrictions and the period following, which consists of the first restrictions period (15 March 2020–14 June 2020), the first gradual restriction lifting (15 June 2020–2 February 2021), the second restrictions period (3 February 2021–27 May 2021), and the second gradual restriction lifting (28 May 2021–31 December 2021). This timeline covers a year prior to the pandemic, approximately a year that includes the start of the pandemic and the imposition of the strictest set of restrictions, and approximately a year where restrictions were imposed.

The data was de-identified and de-duplicated to remove repeat tests and ASTs using the resistance phenotype-based method described by Hindler et al. (2007), such that only the first episode of infection by a given pathogen within a year is kept [38]. An infection type category was added using the orderable name as recorded by the hospital and the specimen type and collection body site. Infection types were categorized into UTIs, RTIs, sterile sites (including bacteremia), and gastrointestinal. 

### 4.4. Microbiological Identification and Antimicrobial Susceptibility Testing (AST) 

The microbiology division of the Department of Pathology and Laboratory Medicine of HMC handles all specimens for microbiological identification and antimicrobial-resistance testing (AST). AST was performed through automated platforms, including BD Phoenix^TM^ (BD Diagnostic Systems, Sparks, MD, USA) and VITEK^®^ (bioMérieux, Marcy-l’Etoile, France), while microbiological identification was performed through MALDI-TOF (Bruker Daltonics, Rheinstetten, Germany). Classification of MRSA as resistant was based on reports from the microbiology laboratory following phenotypic testing, primarily resistance to cefazolin and cloxacillin, while carbapenem-resistant organisms (CROs) were defined as resistant to any of locally tested carbapenems as outlined by the CLSI: meropenem, ertapenem, or meropenem. Species with intrinsic resistance to ertapenem (such as *A. baumannii* and *P. aeruginosa*) were considered CROs if they were resistant to either meropenem or imipenem [39]. MDROs were determined following guidelines defined by Magiorakos et al. (2012) [40].

COVID-19 status and hospitalization dates were also obtained from HMC and added to the data. A bacterial infection was determined to be during a COVID-19 hospitalization if the specimen collection was during hospitalization. Due to the enormous diversity of pathogens, the analysis considered only the critical priority pathogens in the WHO global priority list: antibiotic-resistant bacteria, *A. baumannii*, *P. aeruginosa*, and *Enterobacteriaceae* [15,16]. Additionally, *S. aureus* from the high-priority list was included as it was among the most common pathogens. The remaining high-priority pathogens were responsible for less than 10% of infections each. Additionally, data on the number of international visitors to Qatar and COVID-19 hospitalizations were collected as described previously [13].

### 4.5. Statistical Analysis

All statistical analyses were performed using R (version 4.1.0) [41]. Figures were generated using ggplot 2 (version 3.4.4) and ggpubr version (0.6.0) [42,43]. Descriptive statistics were computed using the gtsummary (version 1.7.0) [44]. Chi-square or Fisher’s exact (depending on the sample size, using the rstatix package version 0.7.2) tests were used to compare the resistance rates across the study period, and post hoc pairwise tests were used to compare the rates between the pandemic periods [45]. Similarly, Chi-square or Fisher tests were used to compare stratification by infection type. The Bonferonni correction was used to correct for multiple testing in all cases. Odds ratios with 95% confidence intervals were computed (using the epitools package version 0.5–10.1) to compare resistance rates between infection episodes that occurred during COVID-19 hospitalization and infection episodes that did not happen during COVID-19 hospitalization (regardless of the patient’s COVID-19 status) [46]. The use of infections that did not occur during COVID-19 hospitalization as a control group is due to the unavailability of hospitalization data in non-COVID-19 patients. Spearman rank correlation was measured (using base R and ggcorrplot version 0.1.4) for the number of antimicrobial-resistant infections and other variables, including the number of international visitors and hospitalized patients [47].

## 5. Conclusions

Despite the fading COVID-19 pandemic causing substantial clinical, social, and economic consequences, there are many lessons to be learned from the evolving infection trends to prepare healthcare across the globe for future pandemic events. Examining the spectrum of specific bacterial pathogens and AMR reported to healthcare during the COVID-19 pandemic in Qatar against imposed travel and social restrictions revealed clear patterns of both infections, described previously in [13], and resistance trends. While specific resistant pathogens such as ESBL-EBC, CIP-SAL, and MRSA showed a significant decrease during the first pandemic restriction period when person-to-person contact was limited. Moreover, ESBL-EBC and MRSA were positively correlated with international travel. On the other hand, CRPA showed a more variable trend with the second restrictions period having a higher rate than others. Additionally, the rate of CRPA was positively correlated with COVID-19 hospitalization. As for CR-EBC, while the overall resistance rate remained consistent throughout the periods, COVID-19 hospitalized patients were more likely to develop infections by them.

While the data presented in the study shows variation in resistance rates and associations with international travel and COVID-19 hospitalization, further investigation is needed. For example, due to data availability, the study used a year before the pandemic as a baseline. Extending the timeline to include more years prior to the pandemic may improve the baseline and better account for the variation over time. Additionally, incorporating information on the hospitalization status and whether the infections were HAIs or CAIs for the non-COVID-19 patients can better delineate whether the association observed for CRPA is due to COVID-19 hospitalization specifically, i.e., the specific conditions associated with COVID-19, or hospitalization in general.

Nevertheless, these findings highlight the potential influence of travel and social gathering in propagating specific resistant pathogens, implying that hospitalized patients with COVID-19 are at risk of different bacterial-resistant profiles. The study’s outcome an help in designing further investigations and building infection prevention and control frameworks at the national level through improved surveillance and emergency response strategies for future pandemics to limit the spread and impact of antibiotic-resistant infections.

## Figures and Tables

**Figure 1 antibiotics-13-00203-f001:**
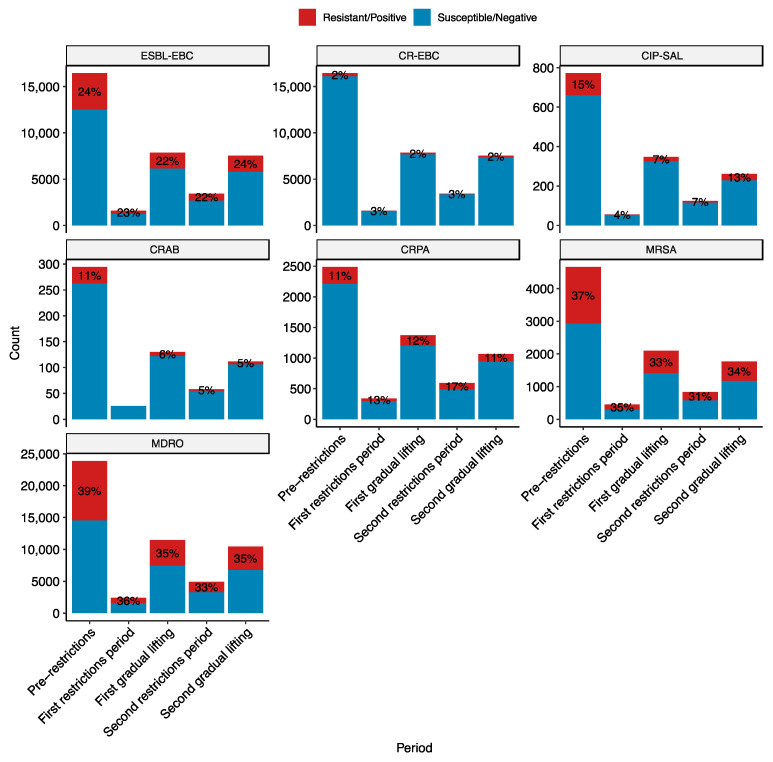
Resistance rates of key bug-drug combinations stratified by the pandemic period. The pandemic periods are pre-restrictions (red; 1 January 2019–14 March 2020), first restrictions (blue; 15 March 2020–14 June 2020). The y-axis is the number of infections by the bug during the period. The red part of the bar and the label correspond to the number and percentage of infections by resistant organisms. The key bug-drug combinations are ESBL-EBC, extended-spectrum β-lactamase producing *Enterobacteriaceae;* CR-EBC, carbapenem-resistant *Enterobacteriaceae*; CIP-SAL, ciprofloxacin-resistant *Salmonella*; CRAB, carbapenem-resistant *Acinetobacter baumannii*; CRPA, carbapenem-resistant *Pseudomonas aeruginosa*; MRSA, methicillin-resistant *Staphylococcus aureus*; and MDRO, multidrug-resistant organisms belonging to the WHO critical and high priority lists. The numbers and the red segments on the bars represent the proportion of resistant infections.

**Figure 2 antibiotics-13-00203-f002:**
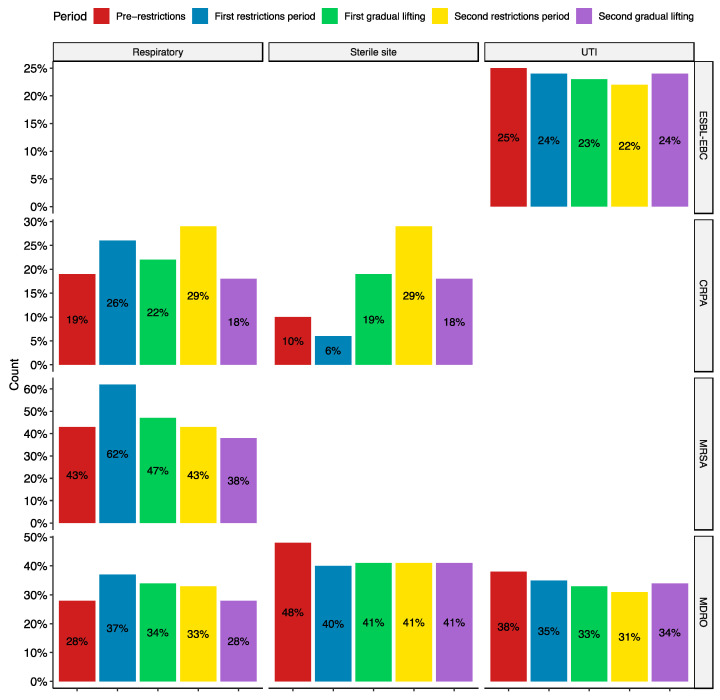
Resistance rates stratified by pandemic period and infection types. All the trends shown are statistically significant (*p* < 0.05). ESBL-EBC, extended-spectrum β-lactamase producing *Enterobacteriaceae;* CRPA, carbapenem-resistant *Pseudomonas aeruginosa*; MRSA, methicillin-resistant *Staphylococcus aureus*; and MDRO, multidrug-resistant organisms belonging to the WHO critical and high priority lists. UTI: urinary tract infections.

**Figure 3 antibiotics-13-00203-f003:**
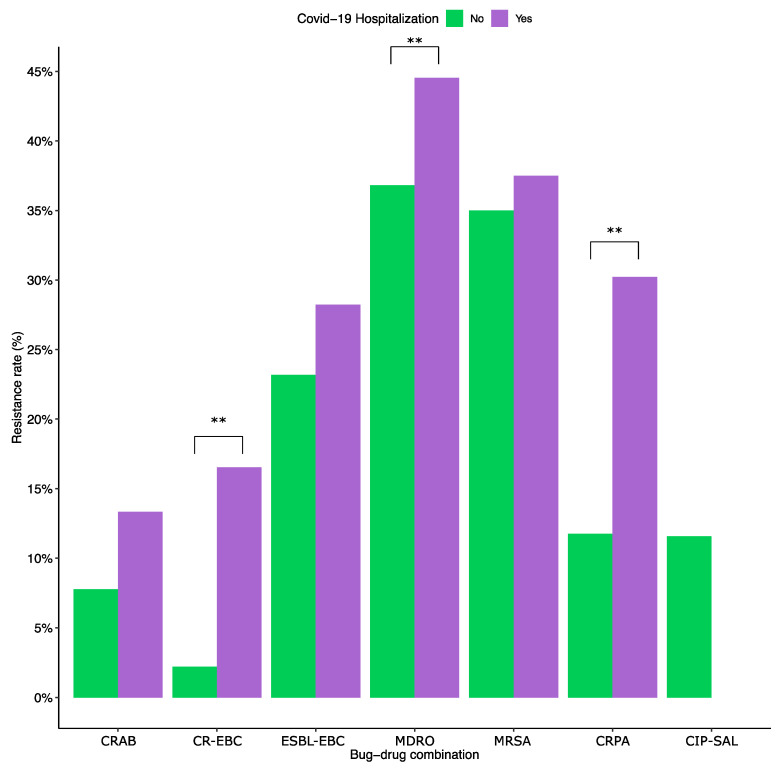
Resistance rates in COVID-19 hospitalized patients vs. non-COVID patients. ESBL-EBC, extended-spectrum β-lactamase producing *Enterobacteriaceae;* CR-EBC, carbapenem-resistant *Enterobacteriaceae*; CIP-SAL, ciprofloxacin-resistant *Salmonella*; CRAB, carbapenem-resistant *Acinetobacter baumannii*; CRPA, carbapenem-resistant *Pseudomonas aeruginosa*; MRSA, methicillin-resistant *Staphylococcus aureus*; and MDR, multidrug-resistant organisms belonging to the WHO critical and high priority lists. The ** indicates statistically significant differences (*p* < 0.01). The odds ratios were 8.85 patients (95% CI: 6.20–12.33; *p* < 0.01) for CR-EBC infections, 3.26 (95% CI: 2.20–4.75; *p* < 0.01) for CRPA, and 1.38 (95% CI: 1.14–1.67; *p* < 0.01) for MDRO. The two groups had no statistically significant difference with the remaining key bug-drug combinations.

**Table 1 antibiotics-13-00203-t001:** Demographic characteristics of the data subset used in the study.

Characteristic	N = 53,183 ^1^
Age	37 (25, 57)
Not reported	13
Sex	
Female	32,088 (60%)
Male	21,092 (40%)
Not reported	3
During COVID-19 Hospitalization	440 (0.8%)
Nationality	
Non-Qatari	38,105 (72%)
Qatari	15,077 (29%)
Not reported	1

^1^ Median (IQR); n (%).

**Table 2 antibiotics-13-00203-t002:** Number of infections caused by species in the WHO priority pathogens for antibiotic research and development stratified by pandemic period.

Organism	Pre-Restrictions **	First Restrictions Period **	First Gradual Lifting **	Second Restrictions Period **	Second Gradual Lifting **	Total
Total	25,394	2510	11,804	5101	10,859	55,668
*Escherichia coli*	10,431	920	4633	1948	4601	22,533
*Staphylococcus aureus*	4657	453	2099	839	1774	9822
*Klebsiella pneumoniae*	3667	408	1961	861	1867	8764
*Pseudomonas aeruginosa*	2484	339	1372	592	1068	5855
Other *Enterobacteriaceae* *	1493	211	897	504	789	3894
*Salmonella*	772	55	348	124	261	1560
*Acinetobacter baumannii*	294	26	130	58	112	620

* Other *Enterobacteriaceae* includes 34 species belonging to 13 genera. ** The dates of the periods are as follows, pre-restrictions: 1 January 2019–14 March 2020, first restrictions period: 15 March 2020–14 June 2020, first gradual lifting: 15 June 2020–2 February 2021, second restrictions period: 3 February 2021–27 May 2021, and second gradual lifting 28 May 2021–31 December 2021.

## Data Availability

Data was obtained from Hamad Medical Corporation and are available upon reasonable request.

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
