# Peer review of "Antimicrobial Resistance in Qatar: Prevalence and Trends before and Amidst the COVID-19 Pandemic"

_antibiotics, 2024, doi:10.3390/antibiotics13030203_

Round 1

Reviewer 1 Report

Comments and Suggestions for Authors

This is a descriptive paper of selected AMR resistance prevalence around the covid pandemic. It is interesting data that will be of interest to the readers.

The main suggestion I have would be to review the conclusions drawn from the data presented. I would suggest having a medical statistician thoroughly review the data set and critically assess what conclusions can be supported. I feel that the generalised conclusions are not supported by the data in its current form.

Do I understand correctly from the manuscript that many of the changes in figure 1 where not significant? I understand that 7 selected differences in line 117 to 127 were significant and the other in figure 1 were not, is this correct?

Specific comments below:

Introduction or discussion needs to have some reference to the expected variance in resistance rates over time. This is needed to inform the reader on the relevance of these results. Specifically, It would be useful to know what changes are typically seen in resistance over time. What is the public health and clinical relevance of a 3-6% increase in resistance over 12 months?

The baseline condition of ~14 months “pre-restrictions” is used to evaluate differences in subsequent time periods of: First restrictions  ~3 months, First gradual lifting ~ 7 months, second restrictions ~ 4 months, second lifting ~ 7 months. As there is no quantification of the variance in the baseline (pre-restrictions) it is not possible to determine if the variations described could be explained by normal variance in the system even in the absence of COVID.

ln 49 “worsened” – use more specific language to clarify what is meant

ln 118, These results should clearly note that this is a percentage point or absolute change in the rate of resistance.  It is worth noting the directions of the change in addition to the percentage points change.

ln 134 “sterile site infections increased during the first” –  I think this should be resistance rates increased rather that infections increased?

ln 135 “These results highlight the changes in the patient population during the pandemic.” This statement needs to be qualified. What specifically does it highlight?

ln 146 ( and methods ln 309) “non-hospitalized patients (regardless of COVID-19 status)” what is this cohort? What are their selection criteria? Presumably this is all out-patients tested positive for a RTI? This seems like an imperfect control group – why was this selected?

ln 226 Is it surprising that patients in hospital with a RTI are more likely to have resistance than outpatients? How do you support there is a link with COVID infection rather than hospitalisation more generally?

Ln 229 Does your data support the statement that “hospitalized COVID-19 patients are more likely to develop secondary hospital and ventilation-associated pneumonia (HAP and VAP) with MDROs such as CR-EBC, and CRPA.”  This is an incomplete/ambiguous comparison, hospitalised COVID patients are more likely to develop HAP/VAP with CR-EBC and CRPA than what?

I believe the data presented here support that hospitalised COVID patients are more likely to have resistant infection, but what evidence presented here supports they “are more likely to develop secondary” HAP/VAP, rather than this secondary infection being the cause of their admission to hospital?

Ln 316 “Examining the spectrum of specific bacterial pathogens and AMR reported to healthcare during the COVID-19 pandemic in Qatar against imposed travel and social restrictions revealed clear patterns of both infections and resistance trends” Where is the data supporting these trends?

Ln 319 “specific resistant pathogens such as ESBL-EBC, CIP-SAL, and MRSA showed downward trends during particular phases of the pandemic restrictions, others such as CRPA and CR-EBC demonstrated an upward trend mainly for hospitalized COVID-19 patients”

Concluding a “downward trends during particular phases of the pandemic restrictions” is misleading to me. Some of these go down and up and many are not significant from my understanding. Additional justification using statistical methods would be needed to

Clarification of what the “particular phases” referred to are would help in providing a clearer conclusion.

Where is the data to support “CRPA and CR-EBC demonstrated an upward trend mainly for hospitalized COVID-19 patients”, what is meant by “mainly for hospitalised patients”?

The statistical analysis methods listed are short and require more detail. There are many comparisons performed here, some discussion of the methods for analysis and how the multiple comparisons were accounted/adjusted for is necessary.

It is also necessary to discuss the relevance of the different time periods. Changes between different periods were declared significant but what is the relevance of this on public health?

Is the single pre-restriction time point sufficient baseline to argue that the small changes seen were in response to COVID rather than expected variance over time?

A critical discussion of the limitations of this study would help the reader.

Ln 325 “The study's outcome should aid in preparing appropriate control and prevention strategies at the national level and tailor guiding directed management” How would these outcomes be useful, some discussion is required here.

Author Response

Reviewer 1:

This is a descriptive paper of selected AMR resistance prevalence around the covid pandemic. It is interesting data that will be of interest to the readers.

The main suggestion I have would be to review the conclusions drawn from the data presented. I would suggest having a medical statistician thoroughly review the data set and critically assess what conclusions can be supported. I feel that the generalised conclusions are not supported by the data in its current form.

Do I understand correctly from the manuscript that many of the changes in figure 1 where not significant? I understand that 7 selected differences in line 117 to 127 were significant and the other in figure 1 were not, is this correct?

Overall comparison showed that significant difference exist with all the selected organisms. Then, a post-hoc analysis was performed to identify where exactly are these differences. Your understainding is correct. The differences listed are the only statistically significant differences.

Specific comments below:

Introduction or discussion needs to have some reference to the expected variance in resistance rates over time. This is needed to inform the reader on the relevance of these results. Specifically, It would be useful to know what changes are typically seen in resistance over time. What is the public health and clinical relevance of a 3-6% increase in resistance over 12 months?

Added sentences to the introduction to establish the global increase in AMR, lines 39-45.

The baseline condition of ~14 months “pre-restrictions” is used to evaluate differences in subsequent time periods of: First restrictions  ~3 months, First gradual lifting ~ 7 months, second restrictions ~ 4 months, second lifting ~ 7 months. As there is no quantification of the variance in the baseline (pre-restrictions) it is not possible to determine if the variations described could be explained by normal variance in the system even in the absence of COVID.

ln 49 “worsened” – use more specific language to clarify what is meant

The sentence has been adjusted to use the more specific “may have contributed to an increase in antimicrobial resistance”.

ln 118, These results should clearly note that this is a percentage point or absolute change in the rate of resistance.  It is worth noting the directions of the change in addition to the percentage points change.

The introductory sentence on line 117 has been changed to explicitly state changes in proportions. Additionally, the direction of the change was explicitly noted in the following sentences where applicable.

ln 134 “sterile site infections increased during the first” –  I think this should be resistance rates increased rather that infections increased?

The sentence was modified accordingly.

ln 135 “These results highlight the changes in the patient population during the pandemic.” This statement needs to be qualified. What specifically does it highlight?

Expanded on the statement to provide more details, lines 141-144.

ln 146 ( and methods ln 309) “non-hospitalized patients (regardless of COVID-19 status)” what is this cohort? What are their selection criteria? Presumably this is all out-patients tested positive for a RTI? This seems like an imperfect control group – why was this selected?

Clarified in lines 160-161 and 345-349. The selection was made due to the unavailability of data on the hospitalization of non-COVID-19 patients. The limitations this poses is also added to the discussion.

ln 226 Is it surprising that patients in hospital with a RTI are more likely to have resistance than outpatients? How do you support there is a link with COVID infection rather than hospitalisation more generally?

Ln 229 Does your data support the statement that “hospitalized COVID-19 patients are more likely to develop secondary hospital and ventilation-associated pneumonia (HAP and VAP) with MDROs such as CR-EBC, and CRPA.”  This is an incomplete/ambiguous comparison, hospitalised COVID patients are more likely to develop HAP/VAP with CR-EBC and CRPA than what?

I believe the data presented here support that hospitalised COVID patients are more likely to have resistant infection, but what evidence presented here supports they “are more likely to develop secondary” HAP/VAP, rather than this secondary infection being the cause of their admission to hospital?

Thank you for raising these important points. We agree that the presented data does not allow for a definite conclusion of increased likelihood within COVID-19 hospitalization vs. hospitalization in general. We agree that while the data supports the higher likelihood of resistant infections, it does not have information to support higher odds in HAP/VAP directly. While this may be the case. The data is incomplete on this aspect. The text has been edited to address these points in lines 264-270 to better state the limitation and clarify the points

Ln 316 “Examining the spectrum of specific bacterial pathogens and AMR reported to healthcare during the COVID-19 pandemic in Qatar against imposed travel and social restrictions revealed clear patterns of both infections and resistance trends” Where is the data supporting these trends?

The infection trends where discussed in a previous publication. A reference was added to point to it. As for the resistance trends, they are discussed in this manuscript.

Ln 319 “specific resistant pathogens such as ESBL-EBC, CIP-SAL, and MRSA showed downward trends during particular phases of the pandemic restrictions, others such as CRPA and CR-EBC demonstrated an upward trend mainly for hospitalized COVID-19 patients” Concluding a “downward trends during particular phases of the pandemic restrictions” is misleading to me. Some of these go down and up and many are not significant from my understanding. Additional justification using statistical methods would be needed to. Clarification of what the “particular phases” referred to are would help in providing a clearer conclusion.Where is the data to support “CRPA and CR-EBC demonstrated an upward trend mainly for hospitalized COVID-19 patients”, what is meant by “mainly for hospitalised patients”?

Modified the conclusion to make the points clearer

The statistical analysis methods listed are short and require more detail. There are many comparisons performed here, some discussion of the methods for analysis and how the multiple comparisons were accounted/adjusted for is necessary.

Added the relevant details to the section (lines 344-359)

It is also necessary to discuss the relevance of the different time periods. Changes between different periods were declared significant but what is the relevance of this on public health?

A discussion of the relevance of the periods and the changes in resistance rates that occurred through them as well as possible factors that influenced these changes is included in the discussions section, particulary lines 226-242 where the effects of the social restrictions are discussed in the context of changes in resistance rates. The discussion is also expanded through the next paragraphs.

Is the single pre-restriction time point sufficient baseline to argue that the small changes seen were in response to COVID rather than expected variance over time?

A critical discussion of the limitations of this study would help the reader.

Added a paragraph in the conclusion to discuss the limitations of the study and address these points.

Ln 325 “The study's outcome should aid in preparing appropriate control and prevention strategies at the national level and tailor guiding directed management” How would these outcomes be useful, some discussion is required here.

Modified the sentence and expanded on it.

Reviewer 2 Report

Comments and Suggestions for Authors

Thanks for giving me an opportunity to review.

If I understand correctly 11.6% of Qatar's population are Qatari citizens. Do you attribute the "healthy immigrant fact" that compared to the overall demographics, Qareatari citizens were over represented in the study.

Table 2: Can you indicate the time periods to make it easier for the readers.

Have you thought of conducting Time Series Analysis. You could try performing interrupted time series if you wish to observe the effect of a certain policy.

Figure 2: 62% of all Staphylococcus Aureus Respiratory samples were resistant to Methicillin. This is a very worrying trend. What factors could be responsible?

Line 115: Is incomplete

Could the Spearman's for all the 4 correlations be provided. Can we get Spearman's correlations for each of the 5 pandemic periods. That would strengthen your claim with no. of infections with no. of international arrivals.

Which R package did you use to calculate the trend?

Author Response

Reviewer 2:

If I understand correctly 11.6% of Qatar's population are Qatari citizens. Do you attribute the "healthy immigrant fact" that compared to the overall demographics, Qareatari citizens were over represented in the study.

This is an excellent point. A comment on this was added in lines 204-211.

Table 2: Can you indicate the time periods to make it easier for the readers.

Added a footnote with the time periods.

Have you thought of conducting Time Series Analysis. You could try performing interrupted time series if you wish to observe the effect of a certain policy.

We appreciate your recommendation for a more comprehensive analysis. However, after careful consideration and given the nature of our research question and dataset, we have opted for a different analytical approach. While we acknowledge the potential insights that time series could offer in future studies.

While I acknowledge the potential insights that time series analysis could off

Figure 2: 62% of all Staphylococcus Aureus Respiratory samples were resistant to Methicillin. This is a very worrying trend. What factors could be responsible?

Thank you for raising this point. 62% of S. aureus being MRSA is concerning. A paragraph discussing the possible contributing factors was added to the discussion, lines 230-239. Additionally, a supporting supplementary figure was added (Supplementary figure S1).

Line 115: Is incomplete

Thank you for noticing this error. The sentence has been completed.

Could the Spearman's for all the 4 correlations be provided. Can we get Spearman's correlations for each of the 5 pandemic periods. That would strengthen your claim with no. of infections with no. of international arrivals.

Added a supplementary table with the Spearman correlations for the four bug-drug combinations with the two variables (international visitors and number of hospitalized COVID-19 patients). Additionally, made some corrections to the spearman correlation values in lines 183 and 185.

Which R package did you use to calculate the trend?

Added information on the packages used in text and in the references.0.1.4

Reviewer 3 Report

Comments and Suggestions for Authors

Authors in this work aimed at describing evolution of bacterial infections and antimicrobial susceptibility patterns in Qatar before and during COVID-19 pandemic.

It is not clear to me whether this study included also outpatient's clinic infections or only hospitalized people, please specify. 

Are there any data about C. difficile infections? This is also a consequences of an increased use of antibiotics and that goes very strict with the antimicrobials resistance issue, especially in elderly (please consider and discuss this doi: 10.3390/antibiotics11020183.).

Also background and discussion can be improved. 

Please consider and use (doi: 10.1089/mdr.2021.0109, and Infez Med2017 Jun 1;25(2):98-107).

Comments on the Quality of English Language

None

Author Response

Authors in this work aimed at describing evolution of bacterial infections and antimicrobial susceptibility patterns in Qatar before and during COVID-19 pandemic.

It is not clear to me whether this study included also outpatient's clinic infections or only hospitalized people, please specify.

The data included all bacterial infections by the selected pathogens. Additionally, a clarification was added to better describe the groups used for the comparison between COVID-19 hospitalized patients and non-COVID-19-hospitalized patients (lines 355-359)

Are there any data about C. difficile infections? This is also a consequences of an increased use of antibiotics and that goes very strict with the antimicrobials resistance issue, especially in elderly (please consider and discuss this doi: 10.3390/antibiotics11020183.).

We agree that C. difficile is a very important pathogen. However, it’s prevalence was low in the dataset. Additionally, we elected to focus on the WHO global priority list for antimicrobial-resistant pathogens.

Also background and discussion can be improved.

The introduction, discussion, and conclusion have been modified to include more informatin/clarifications.

Please consider and use (doi: 10.1089/mdr.2021.0109, and Infez Med. 2017 Jun 1;25(2):98-107).

Thank you for highlighting this publication that gives an example of the impact of stewardship programs on AMR rates. This is a useful topic to address in the introduction of our manuscript and has been addressed accordingly (lines 45-51)

Round 2

Reviewer 1 Report

Comments and Suggestions for Authors

ln 271. Please reconsider the use of the word "develop" here. This that implies that that bacteria infection is secondary or a resultant effect, which is a likely hypothesis, but the data presented here is insufficient to make this claim. This data supports the correlation that resistant infections are more likely to be diagnosed in hospitalised COVID patients but from what data is presented here there is no evidence that these infections "develop" in the patients. It is also a plausible hypothesis that those with existing resistant infections are more likely to be hospitalised with covid.

Author Response

Thank you for raising this point. The infections were considered during COVID-19 hospitalization if the patient was diagnosed after being hospitalized with COVID-19. These COVID-19 patients are admitted to specialized hospitals for COVID-19 treatment. While it may be the case that some of the patients may have been hospitalized prior and then transferred to the COVID facilities, the data was deduplicated to remove recurring tests that showed a positive result with a given pathogen. Thus, a positive test result during COVID hospitalization is a new infection.  However, we agree that the term “develop” may be inaccurate as it suggests a secondary or co-infection. While that may well be the case, the data is sufficient to prove it. That said, more clarification was added to the methods (lines 320-321), and the discussion in line 271 was modified to change “developed” to “be diagnosed with”